# Are Labels Required for Improving Adversarial Robustness?

**Jonathan Uesato**[*]     **Jean-Baptiste Alayrac**[*]     **Po-Sen Huang**[*]

**Robert Stanforth**     **Alhussein Fawzi**     **Pushmeet Kohli**

DeepMind
{juesato,jalayrac,posenhuang}@google.com

## Abstract

Recent work has uncovered the interesting (and somewhat surprising) finding that training models to be invariant to adversarial perturbations requires substantially larger datasets than those required for standard classification. This result is a key hurdle in the deployment of robust machine learning models in many real world applications where labeled data is expensive. Our main insight is that *unlabeled* data can be a competitive alternative to labeled data for training adversarially robust models. Theoretically, we show that in a simple statistical setting, the sample complexity for learning an adversarially robust model from unlabeled data matches the fully supervised case up to constant factors. On standard datasets like CIFAR-10, a simple Unsupervised Adversarial Training (UAT) approach using unlabeled data improves robust accuracy by $21.7\%$ over using 4K supervised examples alone, and captures over $95\%$ of the improvement from the same number of labeled examples. Finally, we report an improvement of $4\%$ over the previous state-of-the-art on CIFAR-10 against the strongest known attack by using additional unlabeled data from the uncurated 80 Million Tiny Images dataset. This demonstrates that our finding extends as well to the more realistic case where unlabeled data is also uncurated, therefore opening a new avenue for improving adversarial training.

## 1 Introduction

Deep learning has revolutionized many areas of research such as natural language processing, speech recognition or computer vision. System based on these techniques are now being developed and deployed for a wide variety of applications, from recommending/ranking content on the web [13, 22] to autonomous driving [7] and even in medical diagnostics [14]. The safety-critical nature of some of these tasks necessitates the need for ensuring that the deployed models are robust and generalize well to all sorts of variations that can occur in the inputs. Yet, it has been shown that the commonly used deep learning models are vulnerable to adversarial perturbations in the input [43], *e.g.* it is possible to fool an image classifier into predicting arbitrary classes by carefully choosing perturbations imperceptible to the human eye.

Since the discovery of these results, many approaches have been developed to prevent this type of behaviour. One of the most effective and popular approaches is known as *supervised adversarial training* [17, 30] which works by generating adversarial samples in an online manner through an inner optimization procedure and then using them to augment the standard training set. Despite

---

[*]Equal contribution, random order.

[†]The authors declare that the present paper is independent of "Unlabeled Data Improves Adversarial Robustness" [10].

substantial work in this space, accuracy of classifiers on adversarial inputs remains much lower than that on normal inputs. Recent theoretical work has offered a reason for this discrepancy, and argues that training models to be invariant to adversarial perturbations requires substantially larger datasets than those required for the standard classification task [41].

This result is a key hurdle in the development and deployment of robust machine learning models in many real world applications where labeled data is expensive. Our central hypothesis is that additional unlabeled examples may suffice for adversarial robustness. Intuitively, this is based on two related observations, explained in Sections 3 and 4.1.2. First, adversarial robustness depends on the smoothness of the classifier around natural images, which can be estimated from unlabeled data. Second, only a relatively small amount of labeled data is needed for standard generalization. Thus, if adversarial training is robust to label noise, labels from supervised examples can be propagated to unsupervised examples to train a smoothed classifier with improved adversarial robustness.

Motivated by this, we explore Unsupervised Adversarial Training (UAT) to use unlabeled data for adversarial training. We study this algorithm in a simple theoretical setting, proposed by [41] to study adversarial generalization. We show that once we are given a single labeled example, the sample complexity of UAT matches the fully supervised case up to constant factors. In independent and concurrent work, [10, 53, 33] also study the use of unlabeled data for improving adversarial robustness, which we discuss in Section 2.

Experimentally, we find strong support for our main hypothesis. On CIFAR-10 and SVHN, with very limited annotated data, our method reaches robust accuracies of 54.1% and 84.4% respectively against the FGSM[20] attack [26]. These numbers represent a significant improvement over purely supervised approaches (32.5% and 66.0%) on same amount of data and almost match methods that have access to full supervision (55.5% and 86.2%), capturing over 95% of their improvement, without labels. Further, we show that we can successfully leverage realistically obtained unsupervised and uncurated data to improve the state-of-the-art on CIFAR-10 at $\varepsilon = 8/255$ from 52.58% to 56.30% against the strongest known attack.

**Contributions. (i)** In Section 3, we propose a simple and theoretically grounded strategy, UAT, to leverage unsupervised data for adversarial training. **(ii)** We provide, in Section 4.1, strong empirical support for our initial hypothesis that *unlabeled data can be competitive with labeled data when it comes to training adversarially robust classifiers*, therefore opening a new avenue for improving adversarial training. **(iii)** Finally, by leveraging noisy and uncurated data obtained from web queries, we set a new state-of-the-art on CIFAR-10 without depending on any additional human labeling.

## 2 Related Work

**Adversarial Robustness.** [5, 43] observed that neural networks which achieve extremely high accuracy on a randomly sampled test set may nonetheless be vulnerable to adversarial examples, or small but highly optimized perturbations of the data which cause misclassification. Since then, many papers have proposed a wide variety of defenses to the so-called adversarial attacks, though few have proven robust against stronger attacks [1, 8, 46]. One of the most successful approaches for obtaining classifiers that are adversarially robust is adversarial training [1, 46]. Adversarial training directly minimizes the adversarial risk by approximately solving an inner maximization problem by projected gradient descent (PGD) to generate small perturbations that increase the prediction error, and uses these perturbed examples for training [17, 26, 30]. The TRADES approach in [54] improves these results by instead minimizing a surrogate loss which upper bounds the adversarial risk. Their objective is very similar to the one used in UAT-OT (UAT with online targets introduced in Section 3.1) but is estimated purely on labeled rather than unlabeled data.

Common to all these approaches, a central challenge is adversarial generalization. For example, on CIFAR-10 with a perturbation of $\varepsilon = 8/255$, the adversarially trained model in [30] achieves an adversarial accuracy of 46%, despite near 100% adversarial accuracy on the train set. For comparison, standard models can achieve natural accuracy of 96% on CIFAR-10 [52]. Several recent papers have studied generalization bounds for adversarial robustness [2, 23, 41, 51]. Of particular relevance to our work, [41] argues that adversarial generalization may require more data than natural generalization. One solution explored in [21] is to use pretraining on ImageNet, a large supervised dataset, to improve adversarial robustness. In this work, we study whether more labeled data is necessary, or whether unlabeled data can suffice. While to our knowledge, this question has not been directly studied, several works such as [20, 40, 42] propose using generative models to detect or denoise adversarial

examples, which can in principle be learned on unlabeled data. However, so far, such approaches have not proven to be robust to strong attacks [1, 46].

**Semi-supervised learning.** Learning from unlabeled data is an active area of research. The semi-supervised learning approach [11] which, in addition to labeled data, also uses unlabeled data to learn better models is particularly relevant to our work. One of the most effective technique for semi-supervised learning is to use smoothness regularization: the model is trained to be invariant to small perturbation applied to unsupervised samples [3, 4, 27, 32, 39, 50]. Of particular relevance to UAT, [32] also uses adversarial perturbations to smooth the model outputs. In addition, co-training [6] and recent extensions [12, 37] use the most confident predictions on unlabeled data to iteratively construct additional labeled training data. These work all focus on improving standard generalization whereas we explore the use of similar ideas in the context of adversarial generalization.

**Semi-supervised learning for adversarial robustness.** The observation that adversarial robustness can be optimized without labels was made independently and concurrently by [10, 33, 53]. Of particular interest, [10] proposes a meta-algorithm Robust Self-Training (RST), similar to UAT. Indeed, the particular instantiation of RST used in [10] and the fixed-target variant of UAT are nearly equivalent: the difference is whether the base algorithm minimizes the robust loss from [54] or the vanilla adversarial training objective [30]. Their results also provide strong, independent evidence that unlabeled and uncurated examples improve robustness on both CIFAR-10 and SVHN.

# 3 Unsupervised Adversarial Training (UAT)

In this section, we introduce and motivate our approach, Unsupervised Adversarial Training (UAT), which enables the use of unlabeled data to train robust classifiers.

**Notation.** Consider the classification problem of learning a predictor $f_\theta$ to map inputs $x \in \mathcal{X}$ to labels $y \in \mathcal{Y}$. In this work, $f$ is of the form: $f_\theta(x) = \arg\max_{y \in \mathcal{Y}} p_\theta(y|x)$, where $p_\theta(.|x)$ is parameterized by a neural network. We assume data points $(x, y)$ are i.i.d. samples from the data-generating joint distribution $P(X, Y)$ over $\mathcal{X} \times \mathcal{Y}$. $P(X)$ denotes the unlabeled distribution over $\mathcal{X}$ obtained by marginalizing out $Y$ in $P(X, Y)$. We assume access to a labeled training set $\mathcal{S}_n = \{(x_i, y_i)\}_{1 \le i \le n}$, where $(x_i, y_i) \sim P(X, Y)$ and an unlabeled training set $\mathcal{U}_m = \{x_i\}_{1 \le i \le m}$, where $x_i \sim P(X)$.

**Evaluation of Adversarial Robustness.** The *natural risk* is $\mathcal{L}_{nat}(\theta) = \mathbb{E}_{(x,y) \sim P(X,Y)} \ell(y, f_\theta(x))$, where $\ell$ is the $0-1$ loss. Our primary objective is minimizing *adversarial risk*: $\mathcal{L}_{adv}(\theta) = \mathbb{E}_{P(X,Y)} \sup_{x' \in N_\epsilon(x)} \ell(y, f_\theta(x'))$. As is common, the neighborhood $N_\epsilon(x)$ is taken in this work to be the $L_\infty$ ball: $N_\epsilon(x) = \{x' : \|x' - x\|_\infty \le \epsilon\}$. Because the inner maximization cannot be solved exactly, we report the surrogate adversarial risk $\mathcal{L}_g(\theta) = \mathbb{E}_{P(X,Y)} \ell(f_\theta(x'), y)$, where $x' = g(x, y, \theta)$ is an approximate solution to the inner maximization computed by some fixed adversary $g$. Typically, $g$ is (a variant of) projected gradient descent (PGD) with a fixed number of iterations.

## 3.1 Unsupervised Adversarial Training (UAT)

**Motivation.** As discussed in the introduction, a central challenge for adversarial training has been the difficulty of adversarial generalization. Previous work has argued that adversarial generalization may simply require more data than natural generalization. We ask a simple question: is more *labeled* data necessary, or is *unsupervised* data sufficient? This is of particular interest in the common setting where unlabeled examples are dramatically cheaper to acquire than labeled examples ($m \gg n$). For example, for large-scale image classification problems, unlabeled examples can be acquired by scraping images off the web, whereas gathering labeled examples requires hiring human labelers.

We now consider two algorithms to study this question. Both approaches are simple – we emphasize the point that large unlabeled datasets can help bridge the gap between natural and adversarial generalization. Later, in Sections 3.2 and 4, we show that both in a simple theoretical model and empirically, unlabeled data is in fact *competitive* with labeled data. In other words, for a fixed number of additional examples, we observe similar improvements in adversarial robustness regardless of whether or not they are labeled.

**Strategy 1: Unsupervised Adversarial Training with Online Targets (UAT-OT).** We note that adversarial risk can be bounded as $\mathcal{L}_{adv} = \mathcal{L}_{nat} + (\mathcal{L}_{adv} - \mathcal{L}_{nat}) \le \mathcal{L}_{nat} + \mathbb{E}_{P(X,Y)} \sup_{x' \in N_\epsilon(x)} \ell(f_\theta(x'), f_\theta(x))$, similarly to the decomposition in [54]. We refer to the first

term as the *classification* loss, and the second terms as the *smoothness* loss. Even for adversarially trained models, it has been observed that the smoothness loss dominates the classification loss on the test set, suggesting that controlling the smoothness loss is the key to adversarial generalization. For example, the adversarially trained model in [30] achieves natural accuracy of 87% but adversarial accuracy of 46% on CIFAR-10 at $\varepsilon = 8/255$.

Notably, the smoothness loss has no dependence on labels, and thus can be minimized purely through unsupervised data. UAT-OT directly minimizes a differentiable surrogate of the smoothness loss on the unlabeled data. Formally, we use the loss introduced in [32] and also used in [54]

$$\mathcal{L}_{\text{unsup}}^{OT}(\theta) = \underset{x \sim P(X)}{\mathbb{E}} \sup_{x' \in \mathcal{N}_\epsilon(x)} \mathcal{D}(p_{\hat{\theta}}(.|x), p_\theta(.|x')), \tag{1}$$

where $\mathcal{D}$ is the Kullback-Leibler divergence, and $\hat{\theta}$ indicates a fixed copy of the parameters $\theta$ in order to stop the gradients from propagating. While [32], which primarily focuses on natural generalization, uses a single step approximation of the inner maximization, we use an iterative PGD adversary, since prior work indicates strong adversaries are crucial for effective adversarial training [26, 30].

**Strategy 2: Unsupervised Adversarial Training with Fixed Targets (UAT-FT).** This strategy directly leverages the gap between standard generalization and adversarial generalization. The main idea is to first train a *base classifier* for standard generalization on the supervised set $\mathcal{S}_n$. Then, this model is used to estimate labels, hence *fixed targets*, on the unsupervised set $\mathcal{U}_m$. This allows us to employ standard supervised adversarial training using these fixed targets. Formally, it corresponds to using the following loss:

$$\mathcal{L}_{\text{unsup}}^{FT}(\theta) = \underset{x \sim P(X)}{\mathbb{E}} \sup_{x' \in \mathcal{N}_\epsilon(x)} \texttt{xent}(\hat{y}(x), p_\theta(.|x')), \tag{2}$$

where $\texttt{xent}$ is the cross entropy loss and $\hat{y}(x)$ is a *pseudo*-label obtained from a model trained for standard generalization on $\mathcal{S}_n$ alone. Thus, provided a sufficiently large unlabeled dataset, UAT-FT recovers a smoothed version of the base classifier, which matches the predictions of the base classifier on clean data, while maintaining stability of the predictions within local neighborhoods of the data.

**Overall training.** For the overall objective, we use a weighted combination of the supervised loss and the chosen unsupervised loss, controlled by a hyperparameter $\lambda$: $\mathcal{L}(\theta) = \mathcal{L}_{\text{sup}}(\theta) + \lambda \mathcal{L}_{\text{unsup}}(\theta)$. The unsupervised loss can be either $\mathcal{L}_{\text{unsup}}^{OT}$ (UAT-OT), $\mathcal{L}_{\text{unsup}}^{FT}$ (UAT-FT) or both (UAT++). Finally, note that the unsupervised loss can also be used on the samples of the supervised set by simply adding the $x_i$'s of $\mathcal{S}_n$ in $\mathcal{U}_m$. The pseudocode and implmenetation details are described in Appendix A.1.

## 3.2 Theoretical model

To improve our understanding of the effects of unlabeled data, we study the simple setting proposed by [41] to analyze the required sample complexity of adversarial robustness.

**Definition 1** (Gaussian model [41])**.** *Let $\theta^* \in \mathbb{R}^d$ be the per-class mean vector and let $\sigma > 0$ be the variance parameter. Then the $(\theta^*, \sigma)$-Gaussian model is defined by the following distribution over $(x, y) \in \mathbb{R}^d \times \{\pm 1\}$: First, draw a label $y \in \{\pm 1\}$ uniformly at random. Then sample the data point $x \in \mathbb{R}^d$ from $\mathcal{N}(y \cdot \theta^*, \sigma^2 I)$.*

In [41], this setting was chosen to model the empirical observation that adversarial generalization requires more data than natural generalization. They provide an algorithm which achieves fixed, arbitrary (say, 1%) accuracy using a single sample. However, to achieve the same adversarial accuracy, they show that any algorithm requires at least $c_1 \varepsilon^2 \sqrt{d} / \log d$ samples and provide an algorithm requiring $n \geq c_2 \varepsilon^2 \sqrt{d}$ samples, for fixed constants $c_1, c_2$.

Here, we show that this sample complexity can be dramatically improved by replacing labeled examples with unlabeled samples. We first define an analogue of UAT-FT to leverage unlabeled data in this setting. For training an adversarially robust classifier, the algorithm in [41] computes a sample mean of per-point estimates. We straightforwardly adapt this procedure for unlabeled data, as in UAT-FT: we first estimate a base classifier from the labeled examples, then compute a sample mean using fixed targets from this base classifier.

**Definition 2** (Gaussian UAT-FT)**.** *Given $n$ labeled examples $(x_1, y_1), \ldots, (x_n, y_n)$ and $m$ unlabeled examples $x_{n+1}, \ldots, x_{n+m}$, let $\hat{w}_{sup}$ denote the sample mean estimator on labeled examples: $\hat{w}_{sup} =$*

$\sum_{i=1}^{n} y_i x_i$. The UAT-FT estimator is then defined as the sample mean $\hat{w} = \sum_{i=n+1}^{n+m} \hat{y}_i x_i$ where $\hat{y}_i = f_{\hat{w}_{sup}}(x_i)$.

Theorem 1 states that in contrast to the purely supervised setting which requires $O(\sqrt{d} / \log d)$ examples, in the semi-supervised setting, a single labeled example, along with $O(\sqrt{d})$ examples are sufficient to achieve fixed, arbitrary accuracy.

**Theorem 1.** *Consider the $(\theta^*, \sigma)$-Gaussian model with $\|\theta^*\|_2 = \sqrt{d}$ and $\sigma \leq \frac{1}{32} d^{1/4}$. Let $\hat{w}$ be the the UAT-FT estimator as in Definition 2. Then with high probability, for $n = 1$, the linear classifier $f_{\hat{w}}$ has $\ell_\infty^\epsilon$-robust classification error at most $1\%$ if*

$$m \geq c\epsilon^2 \sqrt{d}$$

where $c$ is a fixed, universal constant. The proof is deferred to Appendix G. For ease of comparison, we consider the same Gaussian model parameters $\|\theta^*\|_2$ and $\sigma$ as used in [41]. The sample complexity in Theorem 1 matches the sample complexity of the algorithm provided in [41] up to constant factors, despite using unlabeled rather than labeled examples. We now turn to empirical investigation of whether this result is reflected in practical settings.

# 4 Experiments

In section 4.1, we first investigate our primary question: for adversarial robustness, can unlabeled examples be competitive with labeled examples? These operate in the standard semi-supervised setting where we use a small fraction of the original training set as $\mathcal{S}_n$, and provide varying amounts of the remainder as $\mathcal{U}_m$. After observing high robustness, particularly for UAT-FT and UAT++, we run several controlled experiments in section 4.1.2 to understand why this approach works well. In section 4.2, we explore the robustness of UAT to shift in the distribution $P(X)$. Finally, we use UAT to improve existing state-of-the-art adversarial robustness on CIFAR-10, using the 80 Million Tiny Images dataset as our source of unlabeled data.

## 4.1 Adversarial robustness with few labels

**Experimental setup.** We run experiments on the CIFAR-10 and SVHN datasets, with $L_\infty$ constraints of $\varepsilon = 8/255$ and $\varepsilon = 0.01$ respectively, which are standard for studying adversarial robustness of image classifiers [18, 30, 54, 48]. For adversarial evaluation, we report against 20-step iterative FGSM [26], for consistency with previous state-of-the-art [54]. In our later experiments for Section 4.2.2, we also evaluate against a much stronger attack, MultiTargeted [19], which provides a more accurate proxy for the adversarial risk. As we demonstrate in Appendix E.1, the MultiTargeted attack is significantly stronger than an expensive PGD attack with random restarts, which is in turn significantly stronger than FGSM[20]. We follow previous work [30, 54] for our choices of model architecture, data preprocessing, and hyperparameters, which are detailed in Appendix A.

To study the effect of unlabeled data, we randomly split the existing training set into a small supervised set $\mathcal{S}_n$ and use the remaining $N - n$ training examples as a source of unlabeled data. We also split out 10000 examples from the training set to use as validation, for both CIFAR-10 and SVHN, since neither dataset comes with a validation set. We then study the effect on robust accuracy of increasing $m$, the number of unsupervised samples, across different regimes ($m \approx n$ vs. $m \gg n$).

**Baselines.** We compare results with the two strongest existing supervised approaches, standard adversarial training [30] and TRADES [54], which do not use unsupervised data. We also compare to VAT [32], which was designed for *standard* semi-supervised learning but can be adapted for unsupervised adversarial training as explained in Appendix B. Finally, to compare the benefits of labeled and unlabeled data, we compare to the *supervised oracle*, which represents the best possible performance, where the model is provided the ground-truth label even for samples from $\mathcal{U}_m$.

### 4.1.1 Main results

We first test the hypothesis that for adversarial robustness, additional unlabeled data is competitive with additional labeled data. Figure 1 summarizes the results. We report the adversarial accuracy for varying $m$, when $n$ is fixed to $4000$ and $1000$, for CIFAR-10 and SVHN respectively.

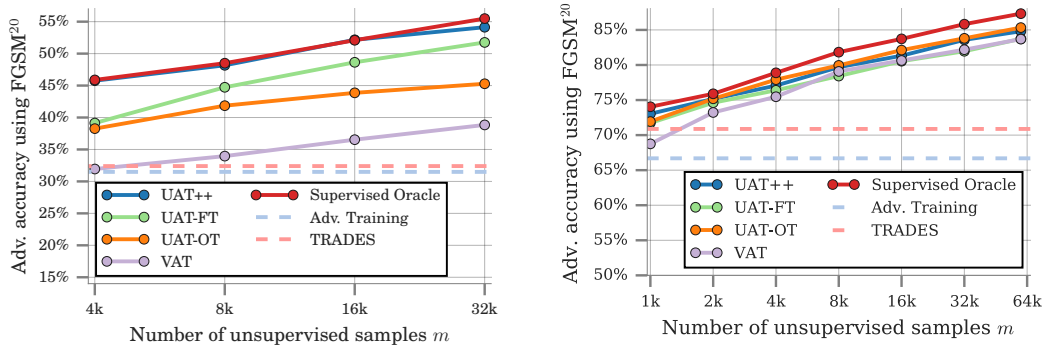

Figure 1: Comparison of labeled data and unsupervised data for improving adversarial generalization on CIFAR-10 (**left,a**) and SVHN (**right,b**)

**Comparison to baselines.** All models show significant improvements in adversarial robustness over the baselines for all numbers of unsupervised samples. With the maximum number (32k / 60k) of unlabeled images, even the weakest UAT model, UAT-OT, shows 12.9% / 16.9% improvement over the baselines not leveraging unlabeled data, and 6.4% / 1.6% improvement over VAT on CIFAR-10 and SVHN, respectively.

**Comparison between UAT variants.** We compare the results of 3 different UAT variants: UAT-OT, UAT-FT, and UAT++. Comparing UAT-FT and UAT-OT, when there are larger number of unsupervised samples, we observe that the UAT-FT shows a significant improvement compared to UAT-OT on CIFAR-10, UAT-OT performs similarly to UAT-FT on SVHN. With smaller numbers of unsupervised samples, the two approaches perform similarly. Empirically, we observe that UAT++, which combines the two approaches, outperforms either individually. We thus primarily use UAT++ for our later experiments.

**Comparison to the oracle.** Figure 1 provides strong support for our main hypothesis. In particular, we observe that when using large unsupervised datasets, UAT++ performs nearly as well as the supervised oracle. In Fig. 1a, with 32K unlabeled examples, UAT++ achieves 54.1% on CIFAR-10, which is 1.4% lower than the supervised oracle. Similarly, with 60K unlabeled data, in Fig. 1b, UAT++ achieves 84.4% on SVHN which is 1.8% lower than the supervised oracle.

**Conclusion.** We demonstrate that, leveraging large amounts of unlabeled examples, UAT++ achieves similar adversarial robustness to supervised oracle, which uses label information. In particular, without requiring labels, UAT++ captures over 97.6% / 97.9% of the improvement from 32K / 60K additional examples compared with supervised oracle on CIFAR-10 and SVHN, respectively.

### 4.1.2   Label noise analysis

Given the effectiveness of UAT-FT and UAT++, we perform an ablation study on the impact of label noise on UAT for adversarial robustness.

**Experimental setup.** To do so, we first divide the CIFAR-10 training set into halves, where the first 20K examples are used for training the base classifier and the latter 20K are used to train a UAT model. Of the latter 20K, we treat 4K examples as labeled, as in Section 4.1.1, and the remaining 16K as unlabeled. We consider two different approaches to introducing label noise. For UAT-FT (Correlated), we produce pseudo-labels using the UAT-FT procedure, where the number of training examples used for the base classifier varies between between 500 and 20K. This produces base classifiers with error rates between 7% and 48%. For UAT-FT (Random), we randomly flip the label to a randomly selected incorrect class. The results are shown in Figure 2.

**Analysis.** In Fig. 2a, in the UAT-FT (Random) case, adversarial accuracy is relatively flat between 1% and 20%. Even with 50% of the examples mislabeled, the decrease in robust accuracy is less than 10%. At the highest level of noise, UAT-FT still obtains a 8.0% improvement in robustness accuracy over the strongest baseline which does not exploit unsupervised data. Similarly, in the

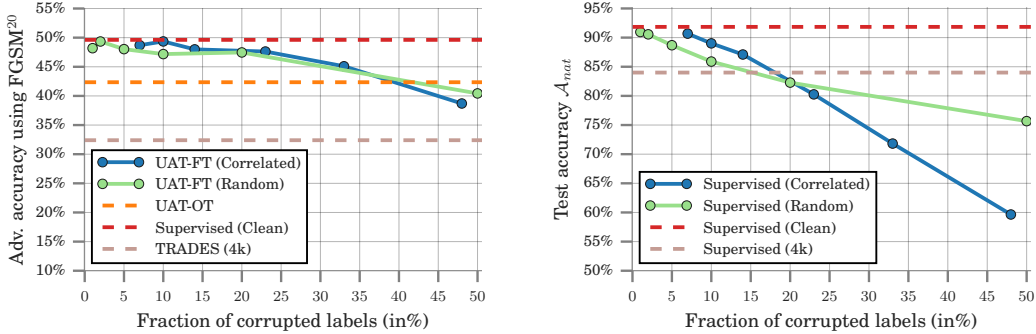

Figure 2: Effects of label noise on adversarial (**left, a**) and natural (**right, b**) accuracies, on CIFAR-10

UAT-FT (Correlated) case, robust accuracy is relatively flat between 7% and 23% noise level, and even at 48% corrupted labels, UAT-FT outperforms the purely supervised baselines by 6.3%.

To understand these results, we believe that the main function of the unsupervised data in UAT is to improve generalization of the smoothness loss, rather than the classification loss. While examples with corrupted labels have limited utility for improving classification accuracy, they can still be leveraged to improve the smoothness loss. This is most obvious in UAT-OT, which has no dependence on the predicted labels (and is thus a flat line in Figure 2a). However, Figure 2a supports the hypothesis that UAT-FT also works similarly, given its effectiveness even in cases where up to half of the labels are corrupted. As mentioned in Section 3, because generalization gap of the classification loss is typically already small, controlling generalization of the smoothness loss is key to improved adversarial robustness.

**Comparison to standard generalization.** We compare the robustness of UAT-FT to label noise, to an analogous pseudo-labeling technique applied to natural generalization. Comparing between Figures 2a and 2b, we observe that with increasing label noise, the rate of degradation in robustness of adversarial trained models is much lower than the rate of degradation in accuracy of models obtained with standard training. In particular, while standard training procedures can be robust to random label noise, as observed in previous work [36, 38], accuracy decreases almost one-to-one (slope -0.78) with correlated errors. This is natural, as with a very large unsupervised dataset, we expect to recover the base classifier (modulo the 4k additional supervised examples).

**Conclusion.** UAT shows significant robustness to label noise, achieving an 8.0% improvement over the baseline even with nearly 50% error in the base classifier. We hypothesize that this is primarily because UAT operates primarily on the smoothness loss, rather than the classification loss, and is thus less dependent on the pseudo-labels.

## 4.2 Unsupervised data with distribution shift

**Motivation.** In Section 4.1, we studied the standard semi-supervised setting, where $P(X)$ is the marginal of the joint distribution $P(X, Y)$. As pointed out in [35], real-world unlabeled datasets may involve varying degrees of distribution shift from the labeled distribution. For example, images from CIFAR-10, even without labels, required human curation to not only restrict to images of the choosen 10 classes but also to ensure that selected images were photo-realistic (line drawings were rejected) or that only one instance of the object was present (see Appendix C of [25] for the full labeler instruction sheet). We thus study whether our approach is robust to such distribution shift, allowing us to fully leverage data which is not only unlabeled, but also uncurated.

We use the **80 Million Tiny Images** [44] dataset (hereafter, 80m) as our uncurated data source, a large dataset obtained by web queries for 75,062 words. Because collecting this dataset required no human filtering, it provides a perfect example of uncurated data that is cheaply available at scale. Notably, CIFAR-10 is a human-labeled subset of 80m, which has been restricted to 10 classes.

**Preprocessing.** Because the majority of 80m contains images distinct from the CIFAR-10 classes, we apply an automated filtering technique similar to [50], detailed in Appendix C. Briefly, we first restrict

| Method | Sup. Data | Unsup. Data | Network | $\mathcal{A}_{nat}$ | $\mathcal{A}_{FGSM^{20}}$ | $\mathcal{A}_{MultiTar.}$ |
|---|---|---|---|---|---|---|
| [48] | CIFAR-10 | ✗ | - | 27.07% | 23.54% | - |
| AT [30] | CIFAR-10 | ✗ | WRN-28 | 87.30% | 47.04% | 44.54% |
| [55] | CIFAR-10 | ✗ | - | 94.64% | 0.15% | - |
| [26] | CIFAR-10 | ✗ | - | 85.25% | 45.89% | - |
| [21] | ImageNet + CIFAR-10 | ✗ | WRN-28 | 87.1% | 57.40% | ≤52.9%* |
| AT-Reimpl. [30] | CIFAR-10 | ✗ | WRN-34 | 87.08% | 52.93% | 47.10% |
| TRADES [54] | CIFAR-10 | ✗ | WRN-34 | 84.92% | 57.11% | 52.58% |
| UAT++ | CIFAR-10 | 80m@100K | WRN-34 | 86.04% | 59.41% | 52.64% |
| UAT++ | CIFAR-10 | 80m@200K | WRN-34 | 85.85% | **62.18%** | **53.35%** |
| UAT++ | CIFAR-10 | 80m@500K | WRN-34 | 78.34% | 58.04% | 48.99% |
| UAT++ | CIFAR-10 | 80m@200K | WRN-70 | 86.75% | 62.89% | 55.04% |
| UAT++ | CIFAR-10 | 80m@200K | WRN-106 | 86.46% | **63.65%** | **56.30%** |

Table 1: Experimental results using 80m Tiny Images dataset (as a unsupervised data) and CIFAR-10 (as supervised data), where $\mathcal{A}_{nat}$ represents the original test accuracy, $\mathcal{A}_{FGSM^{20}}$ represents the adversarial accuracy under 20 step FGSM, and $\mathcal{A}_{MultiTar.}$ represents the adversarial accuracy under the strong `MultiTargeted` attack. WRN-$k$ denotes the Wide-ResNet with depth $k$. '*' indicates it is from [21] using 100 PGD steps with 1000 random restarts, an attack that we have found to be weaker than the `MultiTargeted` attack.

to images obtained from web queries matching the CIFAR-10 classes, and filter out near duplicates of the CIFAR-10 test set using GIST features [34, 15]. For each class, we rank the images based on the prediction confidence from a WideResNet-28-10 model pretrained on the CIFAR-10 dataset. We then take the top 10k, 20k, or 50k images per class, to create the 80m@100K, 80m@200K, and 80m@500K datasets, respectively.

**Overview.** We first conduct a preliminary study on the impact of distribution shift in a low data regime in Section 4.2.1, and we finally demonstrate how UAT can be used to leverage large scale realistic uncurated data in Section 4.2.2.

### 4.2.1 Preliminary study: Low data regime

To study the effect of having unsupervised data from a different distribution, we repeat the same experimental setup described in Section 4.1.1 where we draw $\mathcal{U}_m$ from 80m@200K rather than CIFAR-10. Results are given in Figure 3. For simplicity, we report our best performing method, UAT++, in both settings: $\mathcal{U}_m \subset$ 80m@200K and $\mathcal{U}_m \subset$ CIFAR-10. First, we observe that when using 32K images from unsupervised data, either UAT++ (80m@200K) or UAT++ (CIFAR-10) outperforms the baseline, TRADES [54], which only uses the 4K supervised examples. Specifically, UAT++ with 80m@200K achieves 48.6% robust accuracy, a 16.2% improvement over TRADES. On the other hand, UAT++ performs substantially better when $\mathcal{U}_m$ is drawn from CIFAR-10 rather than 80m@200K, by a margin of 5.5% with 32K unlabeled examples.

**Conclusion.** While unlabeled data from the same distribution is significantly better than off-distribution unlabeled data, the off-distribution unlabeled data is still much better than no unsupervised data at all. In the next section, we explore scaling up the off-distribution case.

### 4.2.2 Large scale regime

We now study whether uncurated data alone can be leveraged to improve the state-of-the-art for adversarial robustness on CIFAR-10. For these experiments, we use subsets of 80m in conjunction with the full CIFAR-10 training set. Table 1 summarizes the results. We report adversarial accuracies against two

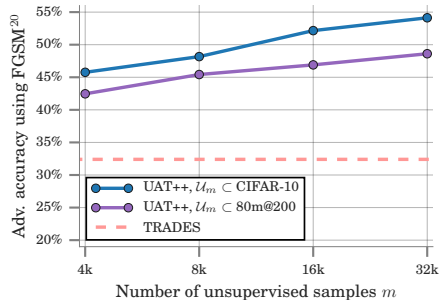

Figure 3: Distribution shift on CIFAR-10

attacks. First, we consider the FGSM [17, 26] attack with 20 steps (FGSM$^{20}$) to allow for direct comparison with previous state-of-the-art [54]. Second, we evaluate against the `MultiTargeted` attack, which we find to be significantly stronger than the commonly used PGD attack with random restarts. Details are provided in Appendix E.1.

**Baselines.** For baseline models, we evaluate the models released by [30, 54]. For fair comparison with our setup, we also reimplement adversarial training (AT-Reimpl. [30]) using the same attack we use for UAT, which we found to be slightly more efficient than the original attack. This is detailed in Appendix A.3. We also compare to [21], which uses more *labeled* data by pretraining on ImageNet. All other reported numbers are taken from [54].

**Comparison with same model.** First, we compare UAT++ with three different sets of unsupervised data (80m@100K, 80m@200K, and 80m@500K) using the same model architecture (WRN-34) as in TRADES. In all cases, we outperform TRADES under FGSM$^{20}$. When using 80m@200K, we improve upon TRADES by 5.07% under FGSM$^{20}$ and 0.77% under the `MultiTargeted` attack. We note the importance of leveraging more unsupervised data when going from 80m@100K to 80m@200K. However, performance degrades when using 80m@500K which we attribute to the fact that 80m@500K contains significantly more out-of-distribution images. Finally, comparing with the recent work of [21], we note that using more unsupervised data can outperform using additional supervised data for pretraining.

**Further analysis.** We run several additional checks against gradient masking [1, 45, 46], detailed in Appendix E. We show that a gradient-free attack, SPSA [46], does not lower accuracy compared to untargeted PGD (Appendix E.2), visualize loss landscapes (Appendix E.3), and empirically analyze attack convergence (Appendix E.4). Overall, we do not find evidence that other attacks could outperform the `MultiTargeted` attack.

**A new state-of-the-art on CIFAR-10.** Finally, when using these significantly larger training sets, we observe significant underfitting, where robust accuracy is low even on the training set. We thus also explore using deeper models. We observe that UAT++ trained on the 80m@200K unsupervised dataset using WRN-106 achieves state-of-the-art performance, +6.54% under FGSM$^{20}$ and +3.72% against `MultiTargeted` attack, compared to TRADES [54]. Our trained model is available on our repository.[1]

## 5  Conclusion

Despite the promise of adversarial training, its reliance on large numbers of labeled examples has presented a major challenge towards developing robust classifiers. In this paper, we hypothesize that annotated data might not be as important as commonly believed for training adversarially robust classifiers. To validate this hypothesis, we introduce two simple UAT approaches which we tested on two standard image classification benchmarks. These experiments reveal that indeed, one can reach near state-of-the-art adversarial robustness with as few as 4K labels for CIFAR-10 (10 times less than the original dataset) and as few as 1K labels for SVHN (100 times less than the original dataset). Further, we demonstrate that our method can also be applied to uncurated data obtained from simple web queries. This approach improves the state-of-the-art on CIFAR-10 by 4% against the strongest known attack. These findings open a new avenue for improving adversarial robustness using unlabeled data. We believe this could be especially important for domains such as medical applications, where robustness is essential and gathering labels is particularly costly [16].

**Acknowledgements.** We would like to especially thank Sven Gowal for helping us evaluate with the `MultiTargeted` attack and for the loss landscape visualizations, as well as insightful discussions throughout this project. We would also like to thank Andrew Zisserman, Catherine Olsson, Chongli Qin, Relja Arandjelović, Sam Smith, Taylan Cemgil, Tom Brown, and Vlad Firoiu for helpful discussions throughout this work.

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
