[Supplementary Material]

# Overview

The appendices are organised as follows. Appendix A provides additional details on the experiments in Section 4. Appendix B details our implementation of VAT [32] adapted for $L_\infty$ adversarial robustness. Appendix C provides additional details on the 80m@N dataset generation procedure. Appendix D details code release. Appendix E includes additional experiments for the adversarial evaluation of our trained models, as well as checks against gradient masking. Finally, we include the proof of Theorem 1 in Appendix G.

# A   Experimental Details

## A.1   Implementation notes

**Model architecture.** For all experiments, we use variants of wide residual networks (WRNs) [52]. In Section 4.1, we use a WRN of width 2 and depth 28 for SVHN and a WRN of width 8 and depth 28 for CIFAR-10. We explore increasing the depth of the network (while keeping width to be 8) to 34, 70 and 106 in Section 4.2.2.

**Data preprocessing.** We use standard data augmentation techniques for images. For CIFAR-10, 4-pixel padding is used before performing random crops of size 32x32 and random left-right flip. For SVHN, 4-pixel padding is also employed before random crops of size 32x32 followed by random color distortions.

**Pseudocode.** We provide pseudocode for our particular implementations of each UAT variant. To simplify notation, when writing $(x, y) \sim \mathcal{U}_m$, the target $y$ is always the fixed target pseudo-label (which is unused in UAT-OT). Recall that these pseudo labels are obtained from a model trained on $\mathcal{S}_n$ alone. $\hat{\mathcal{L}}^{adv}$ and $\hat{\mathcal{L}}^{OT}$ are the empirical estimates of the robust loss from [30] (as in UAT-FT) and $\mathcal{L}^{OT}$ respectively, as defined in the next section.

---

**Algorithm 1** UAT-OT update

---

**Input:** Weight hyperparameter $\lambda$, batch sizes $b_s$ and $b_u$
Sample $b_s$ labeled examples $(\boldsymbol{x}_s, \boldsymbol{y}_s) \sim \mathcal{S}_n$ and $b_u$ unlabeled examples $(\boldsymbol{x}_u, \boldsymbol{y}_u) \sim \mathcal{U}_m$
Compute loss $L = \hat{\mathcal{L}}^{adv}(\boldsymbol{x}_s, \boldsymbol{y}_s; \theta) + \lambda(\frac{b_s}{b_u})\hat{\mathcal{L}}^{OT}(\boldsymbol{x}_u, \boldsymbol{y}_u; \theta)$
Update with gradient $g = \nabla_\theta L$

---

**Algorithm 2** UAT-FT update

---

**Input:** Batch sizes $b_s$ and $b_u$
Sample $b_s$ labeled examples $(\boldsymbol{x}_s, \boldsymbol{y}_s) \sim \mathcal{S}_n$ and $b_u$ unlabeled examples $(\boldsymbol{x}_u, \boldsymbol{y}_u) \sim \mathcal{U}_m$
Merge $\boldsymbol{x} = [\boldsymbol{x}_s; \boldsymbol{x}_u]$; $\boldsymbol{y} = [\boldsymbol{y}_s; \boldsymbol{y}_u]$
Compute loss $L = \hat{\mathcal{L}}^{adv}(\boldsymbol{x}, \boldsymbol{y}; \theta)$
Update with gradient $g = \nabla_\theta L$

---

**Algorithm 3** UAT++ update

---

**Input:** Weight hyperparameter $\lambda$, batch size $b_s$ and $b_u$
Sample $b_s$ labeled examples $(\boldsymbol{x}_s, \boldsymbol{y}_s) \sim \mathcal{S}_n$ and $b_u$ unlabeled examples $(\boldsymbol{x}_u, \boldsymbol{y}_u) \sim \mathcal{U}_m$
Merge $\boldsymbol{x} = [\boldsymbol{x}_s; \boldsymbol{x}_u]$; $\boldsymbol{y} = [\boldsymbol{y}_s; \boldsymbol{y}_u]$
Compute loss $L = \hat{\mathcal{L}}^{adv}(\boldsymbol{x}, \boldsymbol{y}; \theta) + \lambda\hat{\mathcal{L}}^{OT}(\boldsymbol{x}, \boldsymbol{y}; \theta)$
Update with gradient $g = \nabla_\theta L$

---

**Loss implementations.** In the above, the empirical estimates of the losses are defined as follows:

$$\hat{\mathcal{L}}^{adv}(\boldsymbol{x}, \boldsymbol{y}; \theta) = \frac{1}{|(\boldsymbol{x}, \boldsymbol{y})|} \sum_{i=1}^{|(\boldsymbol{x}, \boldsymbol{y})|} \sup_{x'_i \in N_\epsilon(\boldsymbol{x}_i)} \texttt{xent}(\boldsymbol{y}_i, p_\theta(\cdot|x'_i))$$

$$\hat{\mathcal{L}}^{OT}(\boldsymbol{x}, \boldsymbol{y}; \theta) = \frac{1}{|(\boldsymbol{x}, \boldsymbol{y})|} \sum_{i=1}^{|(\boldsymbol{x}, \boldsymbol{y})|} \sup_{x'_i \in N_\epsilon(\boldsymbol{x}_i)} \mathcal{D}(p_{\hat{\theta}}(.|\boldsymbol{x}_i), p_\theta(.|x'_i))$$

We always approximate each maximization with 10-steps of PGD, as described in Appendix A.3. For $\hat{\mathcal{L}}^{OT}$, there are two implementational details of the attack which are not obvious from the pseudocode.

- When computing $\hat{\mathcal{L}}^{OT}$ in UAT-OT, we solve the adversarial optimization with a "hard-label" rather than "soft-label" attack. That is, rather than maximizing the KL directly, we take the hard label $\hat{y} = \arg\max_y p_\theta(y|x)$ and run a PGD attack using $\hat{y}$ as the label. We suspect this is because when $p_\theta(y|x)$ is not completely one-hot, we are maximizing a convex objective. There are (at least) two issues due to this. First, the gradient of the KL, evaluated at $x$, is 0, since $p(y|x)$ is the global minimum. Second, if the random initialization $x'$ causes $p(\hat{y}|x') > p(\hat{y}|x)$, then the gradient will encourage increasing $p(\hat{y}|x')$, rather than decreasing it. While previous work in [54] finds that initializing to a random perturbation of $x$ allows PGD to effectively maximize the KL, using hard labels worked better in our experiments.

- When computing $\hat{\mathcal{L}}^{OT}$ in UAT++, we reuse the adversarial example computed to maximize $\hat{\mathcal{L}}^{adv}$, for computational efficiency.

In both cases, the model loss is still a KL divergence.

**Loss weights.** For all experiments, we used $\lambda = 5$ for both UAT-OT and UAT++. We fixed these values based on early experiments.

**Differences with distribution shift.** For our off-distribution experiment in Section 4.2.2, we make two changes to accomodate distribution shift. First, we compute the loss on the labeled and unlabeled examples using separate forward passes through the network. Without batch norm, this has no effect on the computation. With batch norm, we observe this helps slightly, perhaps because the off-distribution unlabeled data degrades the local batch statistics. Second, we downweight the loss of unlabeled examples by a factor of $b_s/b_u$ (this corresponds to averaging the loss across examples within each separate batch, rather than summing). We suspect that while label noise has little effect, distribution shift degrades performance, and so focusing more on in-distribution examples is helpful.

**Batch sizes.** Throughout Section 4.1, we use batch sizes proportional to the dataset size. In other words, given labeled and unlabeled datasets $\mathcal{S}_n$ and $\mathcal{U}_m$, for a total batch size of $B = 128$, we use $b_s = BN/(N + M)$ and $b_u = BM/(N + M)$. This ensures we perform an equal number of passes through each dataset, and helps avoid overfitting the (small) labeled dataset. For our high data regime experiment in Section 4.2.2, we use fixed $b_s = 512$ and $b_u = 4096$.

**Optimization.** We use the SGD with momentum optimizer for all training jobs. For all training jobs, we use weight decay ($L_2$ regularization) weighted by $5 \times 10^{-4}$.

In Section 4.1.1, we use a total batch size of 128, for $25K$ steps with an initial learning rate of $0.2$ which is decreased by a factor 10 at iterations $15K$, $18K$ and $20K$. This schedule was fine-tuned using the validation set.

In Section 4.2, we use the same learning rate schedule of [54]'s public repository is used for CIFAR-10, *i.e.* initiate with 0.2, then dividing by 10 after 15K and 22K steps.

## A.2 Negative results and observations

We note several observations we made while running experiments. Note that these are much less carefully examined than the results reported in the main paper. Indeed, we suspect that some of these observations may be specific to our specific training setup. However, we include these as we believe they may nonetheless be helpful for future researchers:

- We merge (pseudo)-labeled and unlabeled batches when possible, since this improved robust accuracy, particularly in the small-data regime (when using smaller batch sizes) We suspect this is due to more reliable local batch statistics in batch normalization, since with small batch sizes, the labeled batch can be quite small.

- We found learning rate schedules to be particularly important. For example, in Table 1, the primary difference between our reimplementation of adversarial training, and the implementation in [30] is we use the significantly shorter schedule from [54], which improves robust accuracy by ~3%.

- With our current choices for $\lambda$, we notice a strong regularizing effect from $\hat{\mathcal{L}}^{OT}$. While UAT-FT reaches nearly 100% train robust accuracy by the end of training, the same schedule used with UAT-OT or UAT++ stays below 80% train robust accuracy.

- When using $\hat{\mathcal{L}}^{OT}$ on unsupervised data, we noticed it was necessary to either use fixed parameters $\hat{\theta}$ for computing the target prediction, or to also use fixed targets on the unlabeled data. When both these are missing, the model would learn to predict a uniform class distribution for images in the unsupervised dataset. While such models achieve high training accuracy, test set accuracy stays close to random, even on unperturbed images. On labeled data, both versions with and without backpropagating into target predictions work fine, as reported in [54].

- We tried combining UAT with the recent semi-supervised UDA method [50] but did not see significant improvements in adversarial accuracy.

- We experimented with alternative normalization strategies such as InstanceNorm [47] and GroupNorm [49], but observed slightly worse results.

## A.3 Projected gradient descent (PGD) details

We provide additional details on our untargeted PGD attack, which we use for adversarial training. The untargeted attack optimizes the margin loss objective proposed in [9]

$$ J_\theta^{\mathrm{adv}}(x) = Z(x,\theta)_y - \max_{i \neq y} Z(x,\theta)_i \, , $$

where $Z(x,\theta)_i$ denotes the logit for class $i$, predicted by model $\theta$ on input $x$, and $y$ denotes the true label. The margin loss is negative if and only if $x$ is misclassified.

We optimize this objective with projected gradient descent [26, 30] using the Adam optimizer [24]. Consistent with prior work (e.g. [9, 29]), we find these modifications to improve attack convergence speed when compared to using the negative cross-entropy loss or vanilla gradient updates. During training, we perform 10 steps of optimization, as in [54].

## B  Implementation note on VAT baseline implementation

Recall that for UAT-OT, we use the loss introduced in [32] and also used in [54]

$$ \mathcal{L}_{\mathrm{unsup}}^{OT}(\theta) = \mathop{\mathbb{E}}_{x \sim P(X)} \sup_{x' \in \mathcal{N}_\epsilon(x)} \mathcal{D}(p_{\hat{\theta}}(.|x), p_\theta(.|x')), \tag{3} $$

where $\mathcal{D}$ is the Kullback-Leibler divergence, and $\hat{\theta}$ indicates a fixed copy of the parameters $\theta$ in order to stop the gradients from propagating.

In practice, computing the optimal perturbation $x'$ (in $\sup_{x' \in \mathcal{N}_\epsilon(x)}$) cannot be achieved in closed form, hence approximations have to be proposed. In [32], the authors start from a second order Taylor approximation of $\mathcal{D}(p_{\hat{\theta}}(.|x), p_\theta(.|x'))$, then use a combination of finite differences to efficiently approximate the Hessian and power iteration method to find an estimate of the optimal perturbation. Their end goal being *standard generalization*, they observe that only one iteration of the power method is sufficient for good performance. This specific procedure (Hessian approximation and eigenvector estimation though power method) is specifically tailored to the case where the chosen specification for the adversary corresponds to the $L_2$ ball: $N_\epsilon(x) = \{x' : \|x' - x\|_2 \leq \epsilon\}$, whereas we focus on the $L_\infty$ ball type of constraints.

For fair comparison, we adapt their algorithm to the $L_\infty$ ball by replacing the previously mentioned procedure by one step of FGSM with learning rate equal to $\epsilon$.

UAT-OT mainly differs from VAT in the fact that we instead use 10 steps of PGD to estimate $x'$. Looking at Figure 1 in the main paper, UAT-OT clearly outperforms VAT, indicating that when it comes to adversarial generalization, one has to use a stronger adversary to generate the perturbation.

## C  Dataset Details

We detail the procedure used to generate the 80m@N datasets, used for experiments in Section 4.2. To remove exact and near duplicate images, we follow [15] and use the GIST image descriptors [34] provided with the 80 Million Tiny Images dataset [44]. For every image, we compute the $L_2$ distance to its nearest neighbor in the CIFAR-10 test set, as measured in GIST feature space, and remove all images within distance 0.28, totalling roughly one million images. Following this, we manually checked for near duplicates, by random sampling and visualizing $L_2$ nearest neighbors. In our initial version of this paper, we removed only exact duplicates, which produced robust accuracies roughly 0.5 - 2% higher across all experiments.

For filtering, we use a WRN-28-10 [52] model trained on CIFAR-10, which achieves 96.0% accuracy on the CIFAR-10 test set. Following the procedure used for the original CIFAR-10 dataset [25], for each class in CIFAR-10, we use hyponyms of the class name, based on the Wordnet hierarchy [31]. This leaves roughly 2 million images remaining, though the class distribution is highly non-uniform (of these, 1 million correspond to the "dog" class, while just 50000 correspond to "deer" or "frog"). For each image, we record the probability assigned by the pretrained model to the associated class. We restrictto images with associated probability over $0.5$, and take the top $N/10$ images per class. For the 80m@500K dataset, some classes contain less than 50K such examples. In this case, we randomly duplicate examples, to maintain class balance. Finally, we note that although we use a simple procedure here, and did not experiment with alternatives, we believe developing better procedures to exploit uncurated data is an important and underexplored research direction.

## D  Code Release

Example usage of our best performing model (WRN-106 trained with UAT++ on 80m@200K) as well as all the 80m@N datasets we used to train our models can be found on our github repository.[2]

## E  Adversarial Evaluation Details

### E.1  Multitargeted attack evaluation

We evaluate using the `MultiTargeted`attack proposed in [19], which we have found to be significantly stronger than commonly used PGD attacks, at the cost of increased computation. The `MultiTargeted` implementation is equivalent to the untargeted attack described above, but rather than performing a single optimization, instead runs a targeted attack against each possible class, and returns the image which minimizes the untargeted loss. The targeted margin loss is

$$J_\theta^{\mathrm{adv}}(x) = Z(x,\theta)_y - Z(x,\theta)_t \,,$$

where $t$ is the target class. We use 200 steps with 20 random restarts for each class.

We have found the `MultiTargeted` attack to reliably outperform untargeted PGD attacks, which in turn reliably outperform FGSM[20]. In ensuring the strongest results for untargeted PGD, we found that using 200 steps with 20 random restarts slightly outperforms 100 steps with 1000 random restarts, and hence report results using the former. For example, for our strongest model, trained on the 80m@200K dataset, the FGSM[20]adversarial accuracy is 63.65%, the untargeted PGD adversarial accuracy is 61.10%, and the `MultiTargeted` adversarial accuracy is 56.30%.

## E.2 SPSA evaluation

As an additional check against gradient masking [1, 46], we run a gradient-free attack, SPSA [46], against our strongest model, UAT++ trained on the 80m@200K unsupervised dataset. For SPSA, we use a batch size of 8192 with 40 iterations. We obtain 64.9% adversarial accuracy, similar to the 61.1% obtained by untargeted PGD. We further observe in Figure 4 that SPSA and PGD reliably converge to similar loss values, and that SPSA rarely outperforms PGD. This provides additional evidence that the model's strong performance is not due to gradient masking.

Figure 4: **Analysis of gradient masking with SPSA**: We compare the final values of the margin loss across different images, between PGD and SPSA. Each point represents a single image, which is misclassified when $J_\theta^{\mathrm{adv}}(x) < 0$. Overall, we find that SPSA and PGD converge to similarly adversarial perturbations (points close to the line $y = x$). We observe relatively few images where SPSA outperformed PGD (below the line, shown in red). The upward bend in the dots to the left of $-3$ are an artifact due to the fact that we terminate the SPSA attack early, once we find any image with margin loss below $-3$.

## E.3 Loss landscape analysis

As another check against gradient masking, we look at the adversarial loss landscape from our strongest model. We examine the loss surfaces for four images where the `MultiTargeted` attack succeeded, but untargeted PGD attack, with 200 steps and 20 restarts, did not. Figure 5 shows the untargeted adversarial loss (optimized by PGD) around the nominal image from CIFAR-10. In these loss landscapes, we vary the input along a linear space defined by the worse perturbations found by PGD and a random direction. The $u$ and $v$ axes represent the magnitude of the perturbation added in each of these directions respectively and the $z$ axis represents the loss. For figures on the right hand side of Figure 5, they show a top-view of the loss landscape and indicates that a large portion of $L_\infty$ ball around the nominal image pushes the PGD solution towards the right (rather than the bottom). We observe that the loss landscape is rather smooth, which provides (weak) additional evidence that the strong performance is not due to gradient masking.

## E.4 Attack convergence analysis

As another check against gradient masking, we analyzed the convergence of PGD. Figure 6 shows convergence of untargeted PGD across different random restarts for our strongest model, though we also observe similar patterns across other models. We observed that on randomly selected images, PGD quickly converges, and that the final loss values across random restarts are tightly clustered, indicating PGD likely converges to near-optimal perturbations. Figure 6a shows randomly selected images, and is consistent with what we observe across images where `MultiTargeted` and PGD agree. In the fraction of cases where `MultiTargeted` succeeded but PGD did not, we can find evidence of gradient masking on some images through random restarts. In Figure 6b, there are two images where the final loss varies across different random restarts.

## F Additional Experimental Results

$L_2$ **robustness.** We ran several short experiments to ensure our results hold for $L_2$ in addition to $L_\infty$ robustness. We use 4K labeled and 32K unlabeled examples. On CIFAR-10 at $L_2$ radius $\epsilon = 0.87$, which encloses the $L_\infty$ $\epsilon = 4/255$ ball, the purely supervised model achieves 32.7% robust accuracy, the supervised oracle achieves 53.9%, and UAT almost matches this, with 55.2% robust accuracy. This represents a 21% absolute gain from using unlabeled data, which captures over 90% of the oracle improvement, without using additional labels. We observe similar results for $\epsilon = 0.435$, which encloses the $L_\infty$ $\epsilon = 2/255$ ball, of 47.3% / 70.3% / 66.3% for the purely supervised baseline, supervised oracle, and UAT, respectively.

**Number of necessary labels.** To study the minimum number of labels required while maintaining robustness, we also train CIFAR-10 models using fewer labels. In the body of the paper, we report that with 4K labels (and 32K unlabeled examples), UAT achieves 54.1% robust accuracy, compared to 55.5% for the supervised oracle which uses 36K labeled examples. We also trained models using 2K and 1K labels, which yield robust accuracies of 51.9% and 47.7% respectively. Thus, there is some loss of robustness – with 4K labels, UAT almost exactly matches the performance of the supervised oracle, with only a 1.4% gap, whereas with 2K and 1K labels, the gap is larger. However, even in this regime, UAT still achieves significant adversarial robustness.

## G Proof of Theorem 1

### G.1 Preliminaries

We first provide the following two concentration inequalities which we will use to bound our main quantities of interest.

**Lemma 3** (Concentration of $\chi$-squared distribution). *Let $X \sim \mathcal{N}(0, \sigma^2 I_n)$. Then, provided $\alpha^2 > 2n\sigma^2$,*

$$\mathbb{P}(\|X\|^2 \geq \alpha^2) \leq e^{-\alpha^2/(20\sigma^2)}.$$

*Proof.* The result follows from application of Lemma 1 from [28]. $\square$

**Lemma 4.** *Let $X \sim \mathcal{N}(0, \sigma^2 I_m)$ in $\mathbb{R}^m$. Then*

$$\mathbb{P}(\tfrac{1}{m}\|X\|_1 \geq a) \leq 2^m \exp -\frac{ma^2}{2\sigma^2}$$

*Proof.* Forming the Chernoff bound with $t = \frac{ma}{\sigma^2}$, we have:

$$
\begin{aligned}
\mathbb{P}(\tfrac{1}{m}\|X\|_1 \geq a) &\leq \exp -\frac{ma^2}{\sigma^2} \, \mathbb{E}[\exp \frac{a}{\sigma^2}\|X\|_1] \\
&= \exp -\frac{ma^2}{\sigma^2} \left( \mathbb{E}[\exp \frac{a}{\sigma^2}|X_1|] \right)^m \\
&= \exp -\frac{ma^2}{\sigma^2} \left( \exp \frac{a^2}{2\sigma^2}(1 + \operatorname{erf} \frac{a}{\sigma\sqrt{2}}) \right)^m \\
&= \exp -\frac{ma^2}{\sigma^2} \exp \frac{ma^2}{2\sigma^2} \left( (1 + \operatorname{erf} \frac{a}{\sigma\sqrt{2}}) \right)^m \\
&\leq 2^m \exp -\frac{ma^2}{2\sigma^2}
\end{aligned}
$$

$\square$

### G.2 Main Proof

To bound the robustness, there are two main quantities of interest. First, we need to bound the norm of $\overline{z} = \frac{1}{m}\sum_{i=1}^m \hat{y}_i x_i$, which controls the smoothness of the classifier (Lemma 5). Second, we need to bound the inner product $\langle \overline{z}, \theta^* \rangle$, which controls how well the classifier fits the data (Lemma 6).

The main difficulty is that $\sum_{i=1}^m \hat{y}_i x_i$ is not Gaussian distributed. In particular, while $\sum_{i=1}^m y_i x_i$ follows a Gaussian distribution, our quantity of interest does not, due to the dependence of $\hat{y}_i$ on $x_i$.

**Lemma 5.** *Given a $(\theta^\star, \sigma)$ Gaussian model in $\mathbb{R}^d$, let $h : \mathbb{R}^d \to \{-1, +1\}$ be any classifier. If $\overline{z} = \frac{1}{m} \sum_{i=1}^m \hat{y}_i x_i$ is the sample mean vector of $m$ i.i.d. samples based on predicted classes $\hat{y}_i = h(x_i)$, then we have*

$$\mathbb{P}\left( \|\overline{z}\|_2 \geq (1+c)\|\theta^*\|_2 + 2\sigma\sqrt{\frac{d}{m}} \right) \leq e^{-6\sqrt{d}/5},$$

*with $c = \frac{\sqrt{20}\sigma}{\|\theta^*\|} \sqrt{\frac{\sqrt{d}}{m} + \log 2}$.*

*Proof.* We have

$$\left\| \frac{1}{m} \sum_{i=1}^m \hat{y}_i x_i \right\|_2 = \left\| \frac{1}{m} \sum_{i=1}^m \hat{y}_i (y_i \theta^* + z_i) \right\|_2 \leq \frac{\|\theta^*\|}{m} \sum_{i=1}^m \hat{y}_i y_i + \left\| \frac{1}{m} \sum_{i=1}^m \hat{y}_i z_i \right\|_2$$

$$\leq \|\theta^*\| + \left\| \frac{1}{m} \sum_{i=1}^m \hat{y}_i z_i \right\|_2,$$

where $z_i \sim \mathcal{N}(0, \sigma^2 I)$. We therefore have

$$\mathbb{P}\left( \left\| \frac{1}{m} \sum_{i=1}^m \hat{y}_i x_i \right\|_2 \geq t \right) \leq \mathbb{P}\left( \left\| \frac{1}{m} \sum_{i=1}^m \hat{y}_i z_i \right\|_2 \geq t - \|\theta^*\| \right)$$

$$\leq \mathbb{P}\left( \bigcup_{s_1 = \pm 1, \ldots, s_m = \pm 1} \left\| \frac{1}{m} \sum_{i=1}^m s_i z_i \right\|_2 \geq t - \|\theta^*\| \right)$$

$$\leq \sum_s \mathbb{P}\left( \frac{1}{m} \left\| \sum_{i=1}^m s_i z_i \right\|_2 \geq t - \|\theta^*\| \right)$$

Observe that $\sum_{i=1}^m s_i z_i \sim \mathcal{N}(0, m\sigma^2)$. Now, using the concentration of measure result, whenever $t - \|\theta^*\| \geq \sqrt{\frac{2}{m}}\sigma$ we have:

$$\mathbb{P}\left( \left\| \sum_{i=1}^m s_i z_i \right\|_2^2 \geq m^2(t - \|\theta^*\|)^2 \right) \leq e^{-m^2(t - \|\theta\|)^2/(20m\sigma^2)} = e^{-m(t - \|\theta\|)^2/(20\sigma^2)}$$

Hence, we obtain

$$\mathbb{P}\left( \left\| \frac{1}{m} \sum_{i=1}^m \hat{y}_i x_i \right\|_2 \geq t \right) \leq 2^m e^{-m(t - \|\theta^*\|)^2/(20\sigma^2)}.$$

Let $t = (1+c)\|\theta^*\| + 2\sigma\sqrt{\frac{d}{m}}$. Then, we have

$$\mathbb{P}\left( \left\| \frac{1}{m} \sum_{i=1}^m \hat{y}_i x_i \right\|_2 \geq t \right) \leq 2^m e^{-m\left( c\|\theta^*\| + 2\sigma\sqrt{\frac{d}{m}} \right)^2/(20\sigma^2)}$$

$$\leq 2^m e^{-m\left( c^2\|\theta^*\|_2^2/(20\sigma^2) \right)} e^{-m 4\sigma^2 \frac{d}{m}/(20\sigma^2)}$$

$$= 2^m e^{-m\left( c^2\|\theta^*\|_2^2/(20\sigma^2) \right)} e^{-d/5}$$

Now let $c = \frac{\sqrt{20}\sigma}{\|\theta^*\|_2} \sqrt{\frac{\sqrt{d}}{m} + \log 2}$. Then, the above probability is given by

$$2^m e^{-m\left( \frac{\sqrt{d}}{m} + \log 2 \right)} e^{-d/5} = e^{-\sqrt{d}} e^{-d/5} \leq e^{-\sqrt{d}} e^{-\sqrt{d}/5} = e^{-6\sqrt{d}/5}.$$

$\square$

**Lemma 6.** *Under the conditions of Lemma 5, let $p = \mathbb{E}[\mathbb{I}[h(x) = y]]$ denote the accuracy of classifier $h$. Then we have*

$$\mathbb{P}\left(\langle \overline{z}, \theta^* \rangle \leq \left(\tfrac{7}{4}p - 1\right)\|\theta^*\|^2 - \sqrt{2}\|\theta^*\|\sigma\sqrt{\frac{\log(1/\delta)}{m} + \log 2}\right) \leq (0.995)^{mp} + \delta.$$

*Proof.* We write

$$\mathbb{P}\left(\langle \hat{w}, \theta^* \rangle \leq t\right) = \mathbb{P}\left(\langle \frac{1}{m}\sum_{i=1}^{m}\hat{y}_i x_i, \theta^* \rangle \leq t\right) = \mathbb{P}\left(\langle \frac{1}{m}\sum_{i=1}^{m}\hat{y}_i(y_i\theta^* + z_i), \theta^* \rangle \leq t\right),$$

where $z_i$ are $\mathcal{N}(0, \sigma^2 I)$. The expression inside the probability is equal to

$$\frac{\|\theta^*\|_2^2}{m}\sum_{i=1}^{m}\hat{y}_i y_i + \frac{1}{m}\sum_{i=1}^{m}\langle \hat{y}_i z_i, \theta^* \rangle$$

We bound this expression from below with

$$\frac{\|\theta^*\|_2^2}{m}\sum_{i=1}^{m}\hat{y}_i y_i - \frac{1}{m}\sum_{i=1}^{m}|\langle z_i, \theta^* \rangle|$$

(That is, we consider the worst-case scenario where the random variables $\hat{y}_i$ are given by the negative of the sign of $\langle z_i, \theta^* \rangle$). We therefore get that

$$\mathbb{P}\left(\langle \hat{w}, \theta^* \rangle \leq t - t'\right) \leq \mathbb{P}\left(\frac{\|\theta^*\|_2^2}{m}\sum_{i=1}^{m}\hat{y}_i y_i - \frac{1}{m}\sum_{i=1}^{m}|\langle z_i, \theta^* \rangle| \leq t - t'\right)$$

$$\leq \mathbb{P}\left(\frac{\|\theta^*\|_2^2}{m}\sum_{i=1}^{m}\hat{y}_i y_i \leq t\right) + \mathbb{P}\left(\frac{1}{m}\sum_{i=1}^{m}|\langle z_i, \theta^* \rangle| \geq t'\right)$$

$$= \mathbb{P}\left(\frac{2\|\theta^*\|_2^2}{m}\sum_{i=1}^{m}\mathbb{I}[y_i = \hat{y}_i] - \|\theta^*\|_2^2 \leq t\right) + \mathbb{P}\left(\frac{1}{m}\sum_{i=1}^{m}|\langle z_i, \theta^* \rangle| \geq t'\right)$$

We treat the first term. Let $t = \left(\tfrac{7}{4}p - 1\right)\|\theta^*\|^2$. The first probability term is hence given by

$$\mathbb{P}\left(\sum_{i=1}^{m}\mathbb{I}[y_i = \hat{y}_i] \leq \frac{7}{8}mp\right) \leq \exp\left(-\frac{mp}{2 \cdot 8^2}\right) \leq (0.995)^{mp}.$$

using a Chernoff bound.

We have the following concentration bound on the $\ell_1$ norm of the Gaussian vectors $U \sim \mathcal{N}(0, \|\theta^*\|^2\sigma^2)$:

$$\mathbb{P}\left(\frac{1}{m}\|U\|_1 \geq t'\right) \leq 2^m \exp\left(-t'^2 m/(2\|\theta^*\|^2\sigma^2)\right),$$

by applying Lemma 4. We set $t' = \sqrt{2}\|\theta^*\|\sigma\sqrt{\frac{\log(1/\delta)}{m} + \log 2}$, and when plugging in the above formula, we obtain

$$\mathbb{P}\left(\frac{1}{m}\sum_{i=1}^{m}|\langle z_i, \theta^* \rangle| \geq t'\right) \leq \delta.$$

Hence, we obtain the desired bound. $\square$

We now use these results to achieve the final result in Theorem 1. In what follows, we assume:

- $(x_1, y_1), \ldots, (x_m, y_m)$ are drawn i.i.d. from a $(\theta^\star, \sigma)$ Gaussian model in $\mathbb{R}^d$ with mean norm $\|\theta^\star\|_2 = \sqrt{d}$
- $h : \mathbb{R}^d \to \{-1, +1\}$ is a base classifier with accuracy $p > \frac{3}{4}$, where $p = \mathbb{E}[\mathbb{I}[h(x) = y]]$
- $\overline{z} \in \mathbb{R}^d$ is the sample mean vector $\overline{z} = \frac{1}{m}\sum_{i=1}^{m}\hat{y}_i x_i$, where $\hat{y}_i = h(x_i)$

- $\hat{w} \in \mathbb{R}^d$ is the unit vector in the direction of $\overline{z}$, i.e., $\hat{w} = \overline{z}/\|\overline{z}\|_2$
- $c$ denotes the constant in Lemma 5

**Lemma 7.** *Under these assumptions,*

$$\mathbb{P}\left[ \langle \hat{w}, \theta^\star \rangle \leq \frac{\left(\frac{7}{4}p - 1\right)\sqrt{dm} - \sqrt{d + 2m\sigma^2 \log 2}}{(1+c)\sqrt{m} + 2\sigma} \right]$$

*is bounded above by* $\exp(-6\sqrt{d}/5) + (0.995)^{mp} + \exp(-d/2\sigma^2)$.

*Proof.* By Lemma 6, we have

$$\mathbb{P}\left[ \langle \overline{z}, \theta^* \rangle \geq \left(\tfrac{7}{4}p - 1\right)\|\theta\|^2 - \sqrt{2}\|\theta\|\sigma\sqrt{\frac{\log(1/\delta)}{m} + \log 2} \right] \geq 1 - (0.995)^{mp} - \delta.$$

Further, by Lemma 5, we have

$$\mathbb{P}\left( \|\hat{w}\|_2 \leq (1+c)\|\theta\| + 2\sigma\sqrt{\frac{d}{m}} \right) \geq 1 - e^{-6\sqrt{d}/5},$$

Conditioning on both events with $\delta = \exp(-d/2\sigma^2)$, the overall failure probability is bounded by $\exp(-6\sqrt{d}/5) + (0.995)^{mp} + \exp(-d/2\sigma^2)$. Then, we have

$$
\begin{aligned}
\langle \hat{w}, \theta^\star \rangle &= \frac{\langle \overline{z}, \theta^\star \rangle}{\|z\|_2} \\
&\geq \frac{\left(\frac{7}{4}p - 1\right)d - \sqrt{2d}\sigma\sqrt{\frac{\log 1/\delta}{m} + \log 2}}{(1+c)\sqrt{d} + 2\sigma\sqrt{\frac{d}{m}}} \\
&= \frac{\left(\frac{7}{4}p - 1\right)\sqrt{dm} - \sigma\sqrt{2}\sqrt{\log 1/\delta + m \log 2}}{(1+c)\sqrt{m} + 2\sigma} \\
&= \frac{\left(\frac{7}{4}p - 1\right)\sqrt{dm} - \sqrt{d + 2\sigma^2 m \log 2}}{(1+c)\sqrt{m} + 2\sigma}
\end{aligned}
$$

$\square$

For ease of reference, we provide a relevant lemma proved in [41].

**Lemma 8** ([41])**.** *Assume a $(\theta^\star, \sigma)$-Gaussian model. Let $p \geq 1$, $\varepsilon \geq 0$ be robustness parameters, and let $\hat{w}$ be a unit vector such that $\langle \hat{w}, \theta^\star \rangle \geq \varepsilon \|\hat{w}\|_p^*$, where $\|\cdot\|_p^*$ is the dual norm of $\|\cdot\|_p$. Then the linear classifier $f_{\hat{w}}$ has $\ell_p^\varepsilon$-robust classification error at most*

$$\exp\left( -\frac{(\langle \hat{w}, \theta^\star \rangle - \varepsilon \|\hat{w}\|_p^*)^2}{2\sigma^2} \right).$$

**Lemma 9.** *With probability at least $1 - [\exp(-6\sqrt{d}/5) + (0.995)^{mp} + \exp(-d/2\sigma^2)]$, the linear classifier $f_{\hat{w}}$ has $\ell_\infty^\varepsilon$-robust classification error at most $\beta$ if*

$$\varepsilon \leq \frac{1}{\sqrt{d}} \frac{\left(\frac{7}{4}p - 1\right)\sqrt{dm} - \sqrt{d + 2\sigma^2 m \log 2}}{(1+c)\sqrt{m} + 2\sigma} - \sigma\sqrt{\frac{2 \log 1/\beta}{d}}$$

*Proof.* We follow the approach used for Theorem 21 in [41]. Define

$$\alpha = \frac{\left(\frac{7}{4}p - 1\right)\sqrt{dm} - \sqrt{d + 2\sigma^2 m \log 2}}{(1+c)\sqrt{m} + 2\sigma}$$

so that we can rewrite

$$\varepsilon \leq \frac{1}{\sqrt{d}}\alpha - \sigma\sqrt{\frac{2 \log 1/\beta}{d}}$$

By Lemma 7, we have that $\langle \hat{w}, \theta^\star \rangle \geq \alpha$ with probability at least $1 - [\exp(-6\sqrt{d}/5) + (0.995)^{mp} + \exp(-d/2\sigma^2)]$.

$$\exp\left(-\frac{(\langle \hat{w}, \theta^\star \rangle - \varepsilon\sqrt{d})^2}{2\sigma^2}\right).$$

Since

$$\langle \hat{w}, \theta^\star \rangle - \varepsilon\sqrt{d} \geq \alpha - \left(\frac{1}{\sqrt{d}}\alpha - \sigma\sqrt{\frac{2\log 1/\beta}{d}}\right)\sqrt{d} = \sigma\sqrt{2\log 1/\beta},$$

the robust classification error is bounded above by $\beta$, as desired. $\square$

**Lemma 10.** *Assume $\sigma \leq \frac{1}{32}d^{1/4}$ and $p > 0.99$. Then, with probability at least $1 - [\exp(-6\sqrt{d}/5) + (0.995)^{mp} + \exp(-d/2\sigma^2)]$ the linear classifier $f_{\hat{w}}$ has $\ell_p^\varepsilon$-robust classification error at most $0.01$ if*

$$m \geq \begin{cases} 100 & \text{for } \varepsilon \leq \frac{1}{4}d^{-1/4} \\ 256\,\varepsilon^2\sqrt{d} & \text{for } \frac{1}{4}d^{-1/4} \leq \varepsilon \leq \frac{1}{4} \end{cases}.$$

*Proof.* We first apply Lemma 9 which gives a $\ell_\infty^{\varepsilon'}$-robust classification error at most $\beta = 0.01$ for

$$\begin{aligned}
\varepsilon' &= \frac{1}{\sqrt{d}}\frac{\left(\frac{7}{4}p - 1\right)\sqrt{dm} - \sqrt{d + 2\sigma^2 m\log 2}}{(1+c)\sqrt{m} + 2\sigma} - \sigma\sqrt{\frac{2\log 1/\beta}{d}} \\
&\geq \frac{1}{\sqrt{d}}\frac{\left(\frac{7}{4}p - 1\right)\sqrt{dm} - \sqrt{d + 2\sigma^2 m\log 2}}{(1+c)\sqrt{m} + 2\sigma} - \frac{1}{8}d^{-1/4}
\end{aligned}$$

The remainder is simple algebraic manipulation. First, we consider the case where $\varepsilon \leq \frac{1}{4}d^{-1/4}$. Using $m = 100$, we first bound

$$\begin{aligned}
c &= \frac{\sqrt{20}\sigma}{\sqrt{d}}\sqrt{\frac{1}{m}d^{1/2} + \log 2} \\
&\leq \frac{\sqrt{20}}{32}d^{-1/4}\sqrt{\frac{1}{100}d^{1/2} + \log 2 d^{1/2}} \\
&\leq \frac{\sqrt{20}}{32}d^{-1/4}\sqrt{d^{1/2}} \\
&\leq \frac{1}{5}
\end{aligned}$$

The resulting robustness is

$$\begin{aligned}
\varepsilon' &\geq \frac{1}{\sqrt{d}}\frac{\left(\frac{7}{4}p - 1\right)\sqrt{dm} - \sqrt{d + 2\sigma^2 m\log 2}}{(1+c)\sqrt{m} + 2\sigma} - \frac{1}{8}d^{-1/4} \\
&= \frac{\left(\frac{7}{4}p - 1\right)\sqrt{m} - \sqrt{1 + 2\sigma^2 md^{-1/2}\log 2}}{(1+c)\sqrt{m} + 2\sigma} - \frac{1}{8}d^{-1/4} \\
&\geq \frac{\frac{7}{10}\sqrt{100} - \sqrt{1 + 200\sigma^2 d^{-1/2}\log 2}}{\frac{6}{5}\sqrt{100} + 2\sigma} - \frac{1}{8}d^{-1/4} \\
&\geq \frac{7 - \sqrt{1 + \frac{200}{32^2}\log 2}}{12 + \frac{1}{16}d^{1/4}} - \frac{1}{8}d^{-1/4} \\
&\geq \frac{7 - \sqrt{1 + \frac{200}{32^2}\log 2}}{\left(12 + \frac{1}{16}\right)d^{1/4}} - \frac{1}{8}d^{-1/4} \\
&\geq \frac{1}{4}d^{-1/4} \\
&\geq \varepsilon
\end{aligned}$$

Next, we consider the case where $\frac{1}{4}d^{-1/4} \le \varepsilon \le \frac{1}{4}$. We again bound $c$:

$$
\begin{aligned}
c &= \frac{\sqrt{20}\sigma}{\sqrt{d}}\sqrt{\frac{\sqrt{d}}{m} + \log 2} \\
&\le \frac{\sqrt{20}}{32}d^{-1/4}\sqrt{\frac{1}{16^2\varepsilon^2} + \log 2} \\
&\le \frac{\sqrt{20}}{32}d^{-1/4}\sqrt{\frac{4^2\sqrt{d}}{16^2} + \log 2} \\
&\le \frac{\sqrt{20}}{32}\sqrt{\frac{4^2}{16^2} + \frac{\log 2}{\sqrt{d}}} \\
&\le \frac{1}{5}
\end{aligned}
$$

The resulting robustness is

$$
\begin{aligned}
\varepsilon' &\ge \frac{1}{\sqrt{d}}\frac{\left(\frac{7}{4}p-1\right)\sqrt{dm} - \sqrt{d + 2\sigma^2 m\log 2}}{(1+c)\sqrt{m} + 2\sigma} - \frac{1}{8}d^{-1/4} \\
&\ge \frac{1}{\sqrt{d}}\frac{\left(\frac{7}{4}p-1\right)\sqrt{16^2\varepsilon^2 d^{3/2}} - \sqrt{d + 2\frac{1}{32^2}(16^2\varepsilon^2)d\log 2}}{16(1+c)\varepsilon d^{1/4} + \frac{1}{16}d^{1/4}} - \frac{1}{2}\varepsilon \\
&= \frac{\left(\frac{7}{4}p-1\right)(16\varepsilon) - \sqrt{d^{-1/2} + \frac{16^2}{32^2}\frac{\log 2}{2}d^{-1/2}\varepsilon^2}}{16(1+c)\varepsilon + \frac{1}{16}} - \frac{1}{2}\varepsilon \\
&\ge \frac{\left(\frac{7}{4}p-1\right)(16\varepsilon) - \sqrt{d^{-1/2} + \frac{16^2}{32^24^2}\frac{\log 2}{2}d^{-1/2}}}{4(1+c) + \frac{1}{16}} - \frac{1}{2}\varepsilon \\
&\ge \frac{11.5\varepsilon - 4\varepsilon\sqrt{1 + \frac{\log 2}{128}}}{4\frac{6}{5} + \frac{1}{16}} - \frac{1}{2}\varepsilon \\
&\ge \varepsilon
\end{aligned}
$$

as desired. $\qquad\square$

**Corollary 11.** *Let $(x_0, y_0)$ and $(x_1, y_1), \ldots, (x_m, y_m)$ be drawn i.i.d. from a $(\theta^\star, \sigma)$ Gaussian model with corruption parameter $p$ and mean norm $\sqrt{d}$. Let $\hat{w}_{sup} = y_0 x_0$. Let $\overline{z} \in \mathbb{R}^d$ be the sample mean $\overline{z} = \frac{1}{m}\sum_{i=1}^m \hat{y}_i x_i$, where $\hat{y}_i = f_{\hat{w}_{sup}}(x_i)$. Let the UAT-FT estimator $\hat{w} \in \mathbb{R}^d$ be the unit vector in the direction of $\overline{z}$, i.e., $\hat{w} = \overline{z}/\|\overline{z}\|_2$. Assume $\sigma \le \frac{1}{32}d^{1/4}$. Then, with probability at least $1 - [\exp(-\frac{6\sqrt{d}}{5}) + (0.996)^m + \exp(-\frac{d}{2\sigma^2}) - 2\exp(-\frac{d}{8\sigma^2+1})]$, the linear classifier $f_{\hat{w}}$ has $\ell_p^\varepsilon$-robust classification error at most $0.01$ if*

$$
m \ge \begin{cases} 100 & \text{for } \varepsilon \le \frac{1}{4}d^{-1/4} \\ 256\,\varepsilon^2\sqrt{d} & \text{for } \frac{1}{4}d^{-1/4} \le \varepsilon \le \frac{1}{4} \end{cases}.
$$

Using the given restriction on $\sigma$, we can invoke Corollary 19 from [41] with $\beta = 0.01$. Thus, with probability at least $1 - 2\exp(-\frac{d}{8\sigma^2+1})$, the classification error $p$ of the base classifier $f_{\hat{w}_{sup}}$ is less than $0.01$. Conditioning on this event, we can invoke Lemma 10 with $m$ as given, which yields a robust classification error of $f_{\hat{w}}$ of at most $0.01$, with probability at least $1 - [\exp(-\frac{6\sqrt{d}}{5}) + (0.996)^m + \exp(-\frac{d}{2\sigma^2})]$.

A union bound gives the desired total failure probability.

Figure 5: Adversarial loss landscapes around the nominal images. It is generated by varying the input to the model, starting from the original input image toward either the worst attack found using PGD ($u$ direction) or the one found using a random direction ($v$ direction). For the figures on the left hand side, the z axis represents the loss. For both panels, the diamond-shape represents the projected $L_\infty$ ball of size $\epsilon = 8/255$ around the nominal image.

(a) Convergence on randomly sampled images

(b) Convergence on images where `MultiTargeted` attack succeeded but untargeted PGD did not

Figure 6: **Convergence of PGD.** Each plot shows the convergence of the adversary loss on the same image, across 20 random restarts. On randomly sampled images (top), the loss converges to tightly clustered values. On images where PGD did not find optimal perturbations (bottom), we observe variation in perturbation strength across different restarts for two images in the bottom row.

## Footnotes

[2]https://github.com/deepmind/deepmind-research/tree/master/unsupervised_adversarial_training