[Reviews · NeurIPS 2019]

Reviewer 1



This paper proposes, based on a theoretical analysis of a simple statistical setting, two strategies for improving adversarial generalization via unlabeled data. The method presented, UAT++, attempts to combine these two strategies and improves adversarial generalization significantly using only additional unlabeled data. Overall, the paper is well-written, easy to follow, and has solid yet simple theoretical grounding. The experimental analysis is thorough and all confounding factors (e.g. network architecture) are correctly ablated. The paper compares to a variety of state-of-the-art methods to prove that unsupervised data can indeed improve adversarial robustness. The paper could be improved by addressing what happens to UAT-OT in the Gaussian setting of Schmidt et al (either via a Theorem or intuitive explanation, or even an explanation of why the algorithm is harder to analyze). It would also be interesting to see whether the same algorithm can also improve L2 robustness (one would guess so, but if the phenomenon is limited to l-infinity this would be independently interesting). Minor comments: - [1] and [2] are the same citation - The graphs are somewhat hard to read (e.g. the red is hard to tell apart from the orange after printing, and the violet lines are rather faint). Using a different colorscheme/formatting would improve the readability.

Reviewer 2



originality: The regularization term has been developed before in semi-supervised learning on unlabeled data [26] and in adversarial training as a smoothness term [41]. This paper combines the two and conducts adversarial training by applying the regularization term on unlabeled data. The theoretical and empirical analysis are new. quality: The paper is technically sound overall. However, some crucial points are not well addressed. For example, (1) the proper number of unlabeled data m. It is observed that the proposed method (UAT++) behaves differently on CIFAR10 and SVHN (Fig.1), wrt m. On CIFAR10, it outperforms others starting from m>=4k. On the other hand, on SVHN, it performs worse than even the baseline with a small m and performs similar to VAT. Although theorem 1 is provided on the theoretical aspect of m, there is no connections and analysis of it with the empirical observations. (2) In table 2, it is observed that the performance could drop with an increasing m. The explanation that the unsupervised data "contains more out-of-distribution images" renders the argument at the beginning of sec 4.2 less effective ("robust to distribution shift", "fully leverage data which is not only unlabeled but also uncurated"). clarity: The paper is clearly written and well organized. significance: The idea of improving model robustness using unlabeled data is interesting and is likely to inspire more efforts in this direction. The idea has been empirically verified to some extent; however, some crucial aspects as mentioned above might requires more efforts to be better addressed. ================Updates==================== The rebuttal from the authors has addressed my major concerns on the mis-match between theory and empirical results, as well as the claimed robustness against distribution shift v.s. the actual results. I believe the investigations on using unlabeled data for improving adversarial robustness is of great importance and this work makes a useful step towards it.

Reviewer 3



Quality and Clarity - the main idea is good and overall the paper is interesting. In my opinion, some parts are not clear enough (for example - subsection 3.1, when elaborating on the strategies). Adding pseudo code might help here. Originality - it's the first time that using unlabeled data are shown to effective in the robust learning regime, both theoretically and empirically. Significance - robust models require more samples in order to generalize. Showing that unlabeled data alleviate this problem is crucial because it is much easier (and cheaper) to collect.

[Author Response · NeurIPS 2019]

**General** We would like to thank all three reviewers for their suggestions – we've made these updates in our internal
version and we believe the updated version is stronger as a result. We note reviewers were positive regarding novelty
and significance, noting that the "theoretical and empirical analysis are new" and that "showing that unlabeled data
alleviate[s] this problem [of robust sample complexity] is crucial because it is much easier (and cheaper) to collect."
Reviewers raised several questions regarding extensions of theoretical and empirical results, the relationship between
theory and practice, and implementation clarity. We have investigated these, and briefly summarize our updates below.

**Algorithm 1** UAT++ update

**Input:** Weight hyperparameter $\lambda$, batch size $b_s$ and $b_u$
Sample $b_s$ labeled examples $(\boldsymbol{x}_s, \boldsymbol{y}_s) \sim \mathcal{S}_n$,
        $b_u$ unlabeled examples $(\boldsymbol{x}_u, \boldsymbol{y}_u) \sim \mathcal{U}_m$
Merge $\boldsymbol{x} = [\boldsymbol{x}_s; \boldsymbol{x}_u]$; $\boldsymbol{y} = [\boldsymbol{y}_s; \boldsymbol{y}_u]$
Compute loss $L = \hat{\mathcal{L}}^{adv}(\boldsymbol{x}, \boldsymbol{y}; \theta) + \lambda \hat{\mathcal{L}}^{OT}(\boldsymbol{x}, \boldsymbol{y}; \theta)$
Update with gradient $g = \nabla_\theta L$

**R1** *Different threat models.* For $L_2$ robustness, we observe similar improvements from UAT, which we've added to the
appendix. For example, on CIFAR-10 at $L_2$ radius $\epsilon = 0.87$, with 4K labeled and 32K unlabeled examples, the purely
supervised model achieves 32.7% robust accuracy, the supervised oracle achieves 53.9%, and UAT almost matches this,
with 55.2%. This represents a 21% absolute gain from using unlabeled data, which captures over 90% of the oracle
improvement, without using labels. We observe similar results for $\epsilon = 0.435$, of 47.3% / 70.3% / 66.3% respectively.

*Theoretical analysis of UAT-OT.* It's a good question. We have focused in the paper on UAT-FT, since it performs
significantly better in our experiments. We are confident a similar result should hold for UAT-OT, using a single label.
The rough intuition is that, with many unlabeled examples, the OT loss ensures that $|\langle \hat{w}, \theta^* \rangle| / \|w\|$ is large, and the
single label is necessary only to determine the sign of $\langle \hat{w}, \theta^* \rangle$. We will add a comment on the analysis of UAT-OT.

**R2** *What is the proper amount of unlabeled data $m$?* We understand this as two questions - how theory explains our
empirical observations, and why VAT outperforms UAT++, on SVHN when $m$ is small. For the first question, the
theory suggests performance should increase monotonically with $m$ - indeed, our experiments validate that unlabeled
data always helps, and that larger $m$ strictly improves performance. Regarding the second, we find UAT++ outperforms
VAT when hyperparameters are properly tuned. In our original SVHN experiments, we directly reused CIFAR-10
hyperparameters. We find it assuring that even with zero hyperparameter tuning, these original results are qualitatively
consistent: using unlabeled data provides vast improvements over labeled data alone, and UAT++ outperforms baselines,
particularly in the $m \gg n$ case we view as important for practical applications. After re-tuning learning rates for all
models, the figure above shows that UAT++ outperforms VAT for all $m$.

*Performance drop with increasing $m$ in Table 1.* You're right that UAT is not robust to arbitrarily out-of-distribution
unlabeled examples, and we've clarified the text accordingly. Qualitatively, we observe greater distribution shift in
80m@500K than in 80m@200K. Our main point is that UAT is moderately robust to distribution shift, sufficient for
us to leverage uncurated data to achieve SOTA robust accuracy. This is indeed only a first step – we're excited to see
future research into more effective ways to leverage uncurated, and further out-of-distribution data.

**R3** *Related literature.* We agree that the three papers mentioned provide useful perspectives on robust sample complexity.
Thanks for mentioning them - we've expanded the discussion of this in the updated paper.

*Pseudocode.* We agree – we've significantly updated our appendix with these details, notably pseudocode, but also
experimental procedures, hyperparameters, ablations, and negative results. We've added a significant section on
pseudocode, which we can't include here for space. Algorithm 1 shows the UAT++ update (the others are similar). To
simplify notation, when writing $(x, y) \sim \mathcal{U}_m$, the target $y$ is always the fixed target pseudo-label. $\hat{\mathcal{L}}^{adv}$ and $\hat{\mathcal{L}}^{OT}$ are the
empirical estimates of the robust loss from Madry et al, 2017 (as in UAT-FT) and $\mathcal{L}^{OT}$ respectively. These loss terms are
also further detailed in our updated appendix.

*How many labels are required?* Short answer: not many. Most papers train with 50K labels on CIFAR-10, but we show
that using UAT allows going from 36K to 4K labels, while maintaining adversarial accuracy – robust accuracy against
FGSM-20 only decreases from 55.5% to 54.1%. Long answer: in the Gaussian model, provided sufficient unlabeled
data, only a single label is necessary. Qualitatively, the theoretical model suggests that UAT only needs sufficient
labels for natural generalization (as opposed to robust generalization), which in the Gaussian model is just a single
label. To study this in practice, we tried pushing to even lower label regimes on CIFAR-10, and still observe significant
adversarial accuracy (now added to appendix): 2K labels yields 51.9%, and 1K yields 47.7% under FGSM-20.

[Meta-Review · NeurIPS 2019]

This paper first shows that additional unlabeled data can significantly lower the generalization gap of robust classifiers under a simple Gaussian model. Then, it presents two unsupervised adversarial training (UAT) methods for learning complex deep networks. Empirical improvements on various datasets are significant. However, this paper heavily overlaps with another paper "Unlabeled Data Improves Adversarial Robustness". As a condition to accepting and including the paper in the proceedings, put the following disclaimer in the footnote on the first page: "The authors declare that the present paper is independent of "Unlabeled Data Improves Adversarial Robustness"."